

# Shift in VEGFA isoform balance towards more angiogenic variants is associated with tumor stage and differentiation of human hepatocellular carcinoma

Mikhail S. Chesnokov[1], Polina A. Khesina[1,2], Darya A. Shavochkina[1],
Inna F. Kustova[1], Leonid M. Dyakov[1], Olga V. Morozova[1],
Nikolai S. Mugue[3], Nikolay E. Kudashkin[4], Ekaterina A. Moroz[4],
Yuri I. Patyutko[4] and Natalia L. Lazarevich[1,2]

[1] Institute of Carcinogenesis, FSBI "N.N. Blokhin National Medical Research Center of Oncology" of the Ministry of Health of the Russian Federation, Moscow, Russian Federation
[2] Biological Faculty, M.V. Lomonosov Moscow State University, Moscow, Russian Federation
[3] N.K. Koltzov Institute of Developmental Biology of Russian Academy of Sciences, Moscow, Russian Federation
[4] Institute of Clinical Oncology, FSBI "N.N. Blokhin National Medical Research Center of Oncology" of the Ministry of Health of the Russian Federation, Moscow, Russian Federation

Corresponding author
Natalia L. Lazarevich,
lazarevich.nl@gmail.com

## ABSTRACT

**Background:** Hepatocellular carcinoma (HCC) is the most common and aggressive type of malignant liver tumor. HCC progression depends significantly on its vascularization and formation of new blood vessels. Vascular endothelial growth factor A (VEGFA) is a crucial regulator of tumor vascularization and components of VEGF-induced cell signaling pathways are important targets of therapeutical drugs that demonstrated the highest efficiency in case of advanced HCC (sorafenib and regorafenib). VEGFA is expressed as a set of isoforms with different functional properties, thus VEGFA isoform expression pattern may affect tumor sensitivity to anti-angiogenic drugs. However, information about VEGFA isoforms expression in HCC is still incomplete and contradictory. The present study aims to quantitatively investigate VEGFA isoform expression aberrations in HCC tissue.

**Methods:** A total of 50 pairs of HCC and non-tumor tissue samples were used to evaluate the VEGFA isoform spectrum using RT-PCR and quantitatively estimate changes in isoform expression using RT-qPCR. Correlations between these changes and tumor clinicopathological characteristics were analyzed.

**Results:** We identified VEGFA-189, VEGFA-165, and VEGFA-121 as predominant isoforms in liver tissue. Anti-angiogenic VEGFA-xxxb variants constituted no more than 5% of all mature VEGFA transcripts detected and their expression was not changed significantly in HCC tissue. We demonstrated for the first time that the least active variant VEGFA-189 is frequently repressed in HCC ($p < 0.001$), while no uniform changes were detected for potent angiogenesis stimulators VEGFA-165 and VEGFA-121. Isoform balance in HCC shifts from VEGFA-189 towards VEGFA-165 or VEGFA-121 in the majority of cases ($p < 0.001$). Changes in fractions, but not expression levels, of VEGFA-189 (decrease) and VEGFA-121 (increase) correlated with advanced Tumor-Node-Metastasis (TNM) and Barcelona Clinic Liver Cancer

(BCLC) tumor stages ($p < 0.05$), VEGFA-189 fraction reduction was also associated with poor tumor differentiation ($p < 0.05$).

**Discussion:** A distinct shift in VEGFA isoform balance towards more pro-angiogenic variants occurs in HCC tissue and may modulate overall impact of VEGFA signaling. We suppose that the ratio between VEGFA isoforms is an important parameter governing HCC angiogenesis that may affect HCC progression and be used for optimizing the strategy of HCC therapy by predicting the response to anti-angiogenic drugs.

## INTRODUCTION

Hepatocellular carcinoma (HCC) is the most common and aggressive form of primary liver tumor and ranks second place in cancer-related mortality rates (*Llovet et al., 2016*). Most HCC patients are diagnosed at advanced stages when the efficacy of existing therapeutic approaches is low and overall prognosis is poor (*Wörns & Galle, 2010*). Identification of specific molecular markers and regulatory mechanisms underlying HCC development is of great clinical importance as it can result in development of novel drugs and strategies for targeted HCC therapy.

Angiogenesis is one of the most critical processes involved in HCC pathogenesis (*Liu et al., 2017*). It is mostly stimulated via the vascular endothelial growth factor (VEGF) signaling pathway activated by secreted VEGF proteins interacting with membrane tyrosine-kinase receptors KDR and FLT-1 (*Ferrara, Gerber & LeCouter, 2003*; *Shen, Hsu & Cheng, 2010*). The VEGF family includes five factors, of which VEGFA is the main driver of angiogenesis (*Ferrara, Gerber & LeCouter, 2003*; *Rapisarda & Melillo, 2012*; *Vempati, Popel & Mac Gabhann, 2014*). Multiple VEGFA isoforms generated through alternative splicing divide into two functional groups. VEGFA-xxxa group (where "xxx" is protein chain length) is pro-angiogenic, with VEGFA-121, VEGFA-165, and VEGFA-189 isoforms being most common; these isoforms differ by presence or absence of exons 6 and 7 (Fig. 1A) (*Ferrara, Gerber & LeCouter, 2003*; *Vempati, Popel & Mac Gabhann, 2014*). VEGFA-xxxb isoforms, discerned by the lack of exon 8a, act as angiogenesis suppressors; the most common variant is VEGFA-165b (Fig. 1A) (*Harper & Bates, 2008*). Since exons 6 and 7 encode heparin-binding domains (HBD) responsible for interaction with extracellular matrix proteins, HBD-containing isoforms like VEGFA-189 are tightly bound to the cell surface. VEGFA-165 is semi-soluble while VEGFA-121 is completely diffusible (*Ferrara, Gerber & LeCouter, 2003*; *Vempati, Popel & Mac Gabhann, 2014*). Functional impact of different VEGFA isoforms is also defined by receptors they interact with. VEGFA-165 is more potent angiogenesis stimulator than VEGFA-121 and VEGFA-189; it acts mainly through KDR-neuropilin-1 complex, which is the primary mediator of VEGF signaling (*Ferrara, Gerber & LeCouter, 2003*). Neuropilin-1 potentiates interactions between KDR and VEGFA-165, but not VEGFA-121 and VEGFA-189, so they

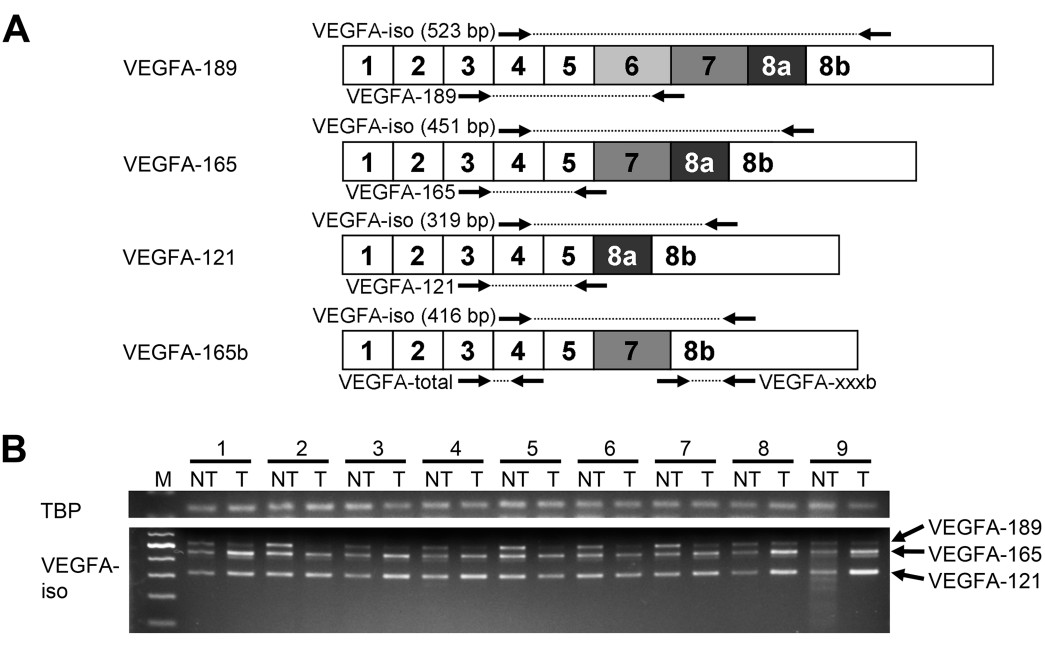

**Figure 1 Spectrum of VEGFA isoforms expressed in HCC and NT liver tissue.** (A) VEGFA isoforms generated through alternative splicing affecting exons 6, 7, and 8. Numbers 1–8 designate exons, arrows indicate primer annealing sites. PCR product lengths are indicated for VEGFA-iso primers, bp—base pairs. (B) Representative results reflecting VEGFA isoforms expression spectrum determined by RT-PCR analysis. M—100 bp DNA marker, NT—non-tumor sample, T—tumor sample. Case numbers are indicated above.

act through much weaker FLT-1 receptor (*Rapisarda & Melillo, 2012*; *Vempati, Popel & Mac Gabhann, 2014*). VEGFA-189 is supposedly the weakest angiogenesis stimulator among three isoforms due to its low solubility and necessity of proteolytic processing (*Plouët et al., 1997*; *Vempati, Popel & Mac Gabhann, 2014*).

High levels of VEGFA expression were reported in different types of malignant tumors including HCC and are associated with advanced stages of disease, poor survival of patients and high recurrence rate (*Ferrara, Gerber & LeCouter, 2003*; *Tseng et al., 2008*; *Shen, Hsu & Cheng, 2010*; *Chekhonin et al., 2013*). A number of VEGFA pathway inhibitors are currently under investigation or already approved for clinical use like bevacizumab, sunitinib, sorafenib, and regorafenib, the last two being the most efficient drugs for therapy of advanced HCC (*Shen, Hsu & Cheng, 2010*; *Rapisarda & Melillo, 2012*; *Bruix et al., 2017*). However, little is known about the functional impact of alternatively spliced forms of VEGFA in hepatocarcinogenesis, while changes in isoform expression pattern may have a considerable effect upon tumor development (*Ladomery, 2013*; *Berasain et al., 2014*; *Elizalde et al., 2014*). The importance of the evaluation of VEGFA variants expression in tumors is clearly demonstrated by investigation of the clinical impact of the anti-angiogenic VEGFA-165b isoform. It is predominant in many normal tissues, while in melanoma, colorectal and bladder cancer the balance shifts towards the highly angiogenic VEGFA-165 variant (*Harper & Bates, 2008*; *Varey et al., 2008*). VEGFA-165b competes with pro-angiogenic isoforms and inhibits VEGFA-165-induced

**Table 1 Clinicopathological characteristics of examined HCC cases.**

| Characteristic | Number of cases ($n = 50$) |
|---|---|
| Age, years (mean ± SD) | 47.8 ± 20.8 |
| Gender, male/female | 27/23 |
| TNM staging, I/II/III/IV | 20/4/15/11 |
| BCLC staging, A/B/C/D | 23/7/20/0 |
| Tumor size, cm (mean ± SD) | 9.8 ± 5.3 |
| Intrahepatic metastases, yes/no | 14/36 |
| Lymph node metastases, yes/no | 9/41 |
| Distant metastases, yes/no | 4/46 |
| Tumor capsule presence, absent/feeble/prominent/N/A[a] | 10/16/17/7 |
| Invasion into blood vessels, yes/no | 21/29 |
| Tumor vascularity[b], low/moderate/high/N/A | 4/10/16/20 |
| Histological differentiation, Edmondson-Steiner grade, G1/G2/G3/G4/Gx[c] | 8/22/6/0/14 |
| Alpha-fetoprotein serum level, low (<50 ng/ml)/high (>50 ng/ml)/N/A | 33/15/2 |
| Ascites, yes/no | 3/47 |
| Cirrhosis, yes/no | 7/43 |
| Tumor necrosis, yes/no/N/A | 30/19/1 |

Notes:
[a] N/A—data not available.
[b] Estimated by average number of visible vessels in histological slides of tumor tissue.
[c] Gx—Edmondson-Steiner grade not applicable.

angiogenesis; VEGFA-165b overexpression or administration of recombinant protein inhibits tumor growth in cancer xenograft models, indicating its potential as an anti-cancer agent (*Harper & Bates, 2008*; *Varey et al., 2008*; *Peiris-Pagès, 2012*). On the other hand, VEGFA-165b binds to the anti-angiogenic drug bevacizumab with equal affinity as VEGFA-165 and thus reduces its efficacy (*Varey et al., 2008*; *Bates et al., 2012*).

Thus, evaluation of VEGFA isoforms expression pattern in tumor is important for a rational choice of anti-angiogenic therapy, which is considered to be a promising approach for HCC treatment. However, no quantitative analysis exploring tumor-specific alterations of VEGFA isoforms expression in HCC has been reported yet. The present study aims to compare the full spectrum of VEGFA isoforms expressed in paired HCC and non-tumorous (NT) liver samples and to quantitatively evaluate changes in the expression of major isoforms and their association with tumor clinicopathological characteristics.

# MATERIALS AND METHODS

## Clinical samples

A total of 50 HCC tissue samples and 50 corresponding NT tissue samples were obtained from patients diagnosed with HCC after tumor resection, fresh-frozen in liquid nitrogen and stored at −70 °C. All patients were hepatitis-negative; additional data on clinicopathological parameters are presented in Table 1. HCC diagnosis and origin of samples (tumor or NT tissue) were confirmed by histopathological analysis performed by two experienced histopathologists specializing in liver cancer. All tumor samples selected

for the study met histological TCGA standards (*Nguyen et al., 2011*) and contained more than 80% tumor nuclei and less than 20% necrotic cells. NT liver samples were taken at least 2 cm away from tumor margin. All procedures performed were in accordance with Declaration of Helsinki (1964) and its later amendments (*World Medical Association, 2013*) or comparable ethical standards and were approved by medical ethics committee of FSBI "N.N. Blokhin National Medical Research Center of Oncology" of the Ministry of Health of the Russian Federation. Written informed consent was obtained from all individual participants included in the study.

## Reverse transcription-polymerase chain reaction analysis of VEGFA expression

Reverse transcription was performed using total RNA isolated from 30 mg of tissue with PureLink RNA Mini Kit (ThermoFisher Scientific, Waltham, MA, USA) using random hexanucleotides and RevertAid Reverse Transcriptase (ThermoFisher Scientific, Waltham, MA, USA).

Semi-quantitative reverse transcription-polymerase chain reaction (RT-PCR) analysis of VEGFA isoforms expression spectrum was performed with VEGFA-iso primers flanking variable mRNA region (Fig. 1A). Primer sequences and RT-PCR conditions are provided in Table S1 and Data S1. TATA-binding protein (TBP) was used as a housekeeping gene. PCR products were analyzed by electrophoresis, purified and verified by sequencing (Data S2; Table S2).

Quantitative RT-qPCR analysis of VEGFA isoforms expression was performed using primers specifically detecting different VEGFA isoforms expressed in liver or isoform groups: VEGFA-total (detects all VEGFA transcripts), VEGFA-xxxb, VEGFA-189, VEGFA-165, VEGFA-121 (Fig. 1A; Tables S1 and S2). The amount of unspliced VEGFA transcripts containing intron 5 was evaluated using VEGFA-intron5 primers (Table S1; Data S3). PCR conditions were experimentally optimized to achieve reaction efficiency of 98–102% (Table S1; Data S1). Transcript abundance was estimated using standard samples containing known quantities of corresponding PCR amplicon copies that were obtained by cloning PCR products into pAL2-T vector using Quick-TA kit (Evrogen, Moscow, Russian Federation) and verified by sequencing (Data S2; Table S2). For each specimen, quantity of VEGFA transcripts was normalized to TBP copy number and changes in relative expression levels and fractions of single isoforms in total pool of VEGFA transcripts were calculated (Data S1).

## Statistical analysis

Differences between observation groups were evaluated using paired sample sign test. Correlations were evaluated using Spearman's rank test, for that numerical clinical parameters (age, tumor size, TNM stage) were used as is, all categorical parameters were assigned rank values (see Table S3 for details). Statistical significance was accepted with $p < 0.05$. When analyzing individual cases, twofold or stronger changes in gene expression were considered significant. Statistical analysis and graph plotting were performed using OriginPro 2016 software (OriginLab Corporation, Northampton, MA, USA).

## RESULTS

### VEGFA-189, VEGFA-165, and VEGFA-121 are major VEGFA isoforms expressed in liver tissue

To explore the full spectrum of VEGFA isoforms expressed in HCC and NT tissue, we performed semi-quantitative RT-PCR analysis using a preliminary panel of 20 HCC cases. We used VEGFA-iso primers that flank the variable region of VEGFA mRNA and amplify several PCR products of different lengths corresponding to certain VEGFA variants (Fig. 1A) The major isoforms expressed in all examined NT and most of HCC specimens were VEGFA-189 (523 bp), VEGFA-165 (451 bp), and VEGFA-121 (319 bp) (Fig. 1; Fig. S1). Isoform identity was verified by sequencing of PCR products (Data S2; Table S2). Decrease in VEGFA-189 level was the predominant aberration of VEGFA isoforms expression found in HCC samples; we also observed occasional up- or downregulation of VEGFA-165 and VEGFA-121 (Fig. 1B). No prominent bands corresponding to other VEGFA variants were detected.

### VEGFA-189 isoform expression is frequently downregulated in HCC

To further explore changes in VEGFA variants expression, we performed quantitative RT-qPCR-analysis of expanded set comprising 50 HCC cases with primers specific to VEGFA isoforms or isoform groups (Tables S1 and S2). Raw data obtained in RT-qPCR analysis for each examined sample are presented in Data S4. VEGFA-total primers were used to evaluate the amount of all VEGFA isoforms expressed in examined samples (Fig. 1A); however, a certain fraction of all VEGFA transcripts retain unspliced intron 5 sequences that can be detected using VEGFA-intron5 primers (Table S2; Data S3). The fraction of unspliced transcripts in examined samples varied from 0.1% to 34.7% (Fig. S2). We therefore calculated the amount of all mature VEGFA transcripts (referred to as "VEGFA-spliced") by subtracting VEGFA-intron5 quantity from that of VEGFA- total (Data S3). Analysis of single cases revealed considerable heterogeneity of VEGFA-spliced changes (both in direction and magnitude), no significant changes in VEGFA-spliced expression were detected between HCC and NT sample sets. Only 11 cases of 50 (22%) displayed more than twofold change of VEGFA-spliced expression (Fig. 2A).

Using primers specific to anti-angiogenic VEGFA-xxxb isoforms, we demonstrated that their fraction in examined samples does not exceed 5% of all spliced VEGFA transcripts. While higher than twofold changes in VEGFA-xxxb expression level are present in part of the examined cases (Fig. S3), there is no significant difference between VEGFA-xxxb expression levels in NT and HCC sample sets (Fig. 2B). Due to these facts, we focused further studies on expression of pro-angiogenic isoforms VEGFA-189, VEGFA-165, and VEGFA-121 using isoform-specific primers.

To verify that our approach to VEGFA isoform quantitation is correct, we compared the sum of the copy numbers obtained for VEGFA-189, VEGFA-165, and VEGFA-121 isoforms with a VEGFA-spliced copy number in every examined sample. The average ratio of these values across the panel was 99.8% (95% confidence interval: 97.1–102.5%)

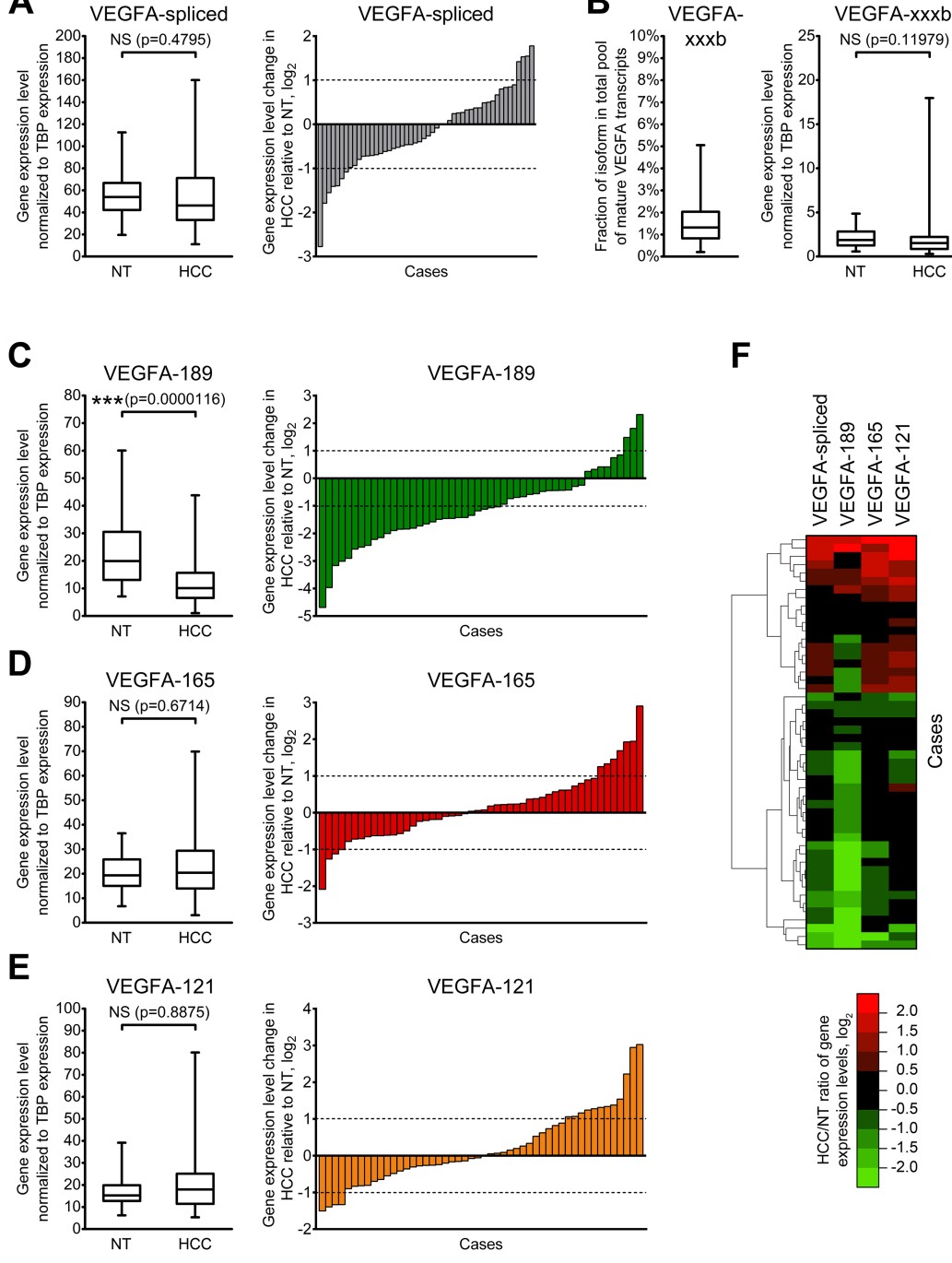

**Figure 2 Alterations of expression levels of VEGFA isoforms in HCC tissue samples in comparison to corresponding NT samples.** (A) Changes in expression level of all mature VEGFA transcripts. (B) VEGFA-xxxb fractions in all examined samples and expression levels in NT and HCC sample sets. (C–E) Changes in expression levels of VEGFA-189 (C), VEGFA-165 (D), and VEGFA-121 (E). (F) Cluster analysis of changes in VEGFA isoforms expression levels. Data for NT or HCC sample sets ($n = 50$) or all samples set ($n = 100$) are presented as box-and-whisker plots, data for individual cases are presented as HCC/NT ratios in logarithmic scale. NS—non-significant difference.

indicating that VEGFA-189, VEGFA-165, and VEGFA-121 are indeed the main components of total VEGFA transcripts pool in examined specimens.

VEGFA-189 is significantly downregulated in HCC sample set in the comparison to the NT sample set, while no significant changes were observed in VEGFA-165 and VEGFA-121 expression (Figs. 2C–2E). Analysis of individual cases confirmed frequent VEGFA-189 repression in HCC ($\geq$2-fold downregulation was detected in 56% cases) while VEGFA-165 and VEGFA-121 variants displayed significant changes in smaller fraction of cases (22% and 32%, respectively) (Figs. 2C–2E). Hierarchical clustering revealed that a decrease in total VEGFA-spliced expression occurs mostly due to VEGFA-189 repression, while its increase is associated with VEGFA-165 and VEGFA-121 upregulation (Fig. 2F).

## Shift of VEGFA isoforms balance occurs in HCC tissue and is associated with tumor clinicopathological characteristics

Since different VEGFA-xxxa isoforms exhibit different angiogenesis-stimulating properties, a shift in their balance can considerably modulate VEGFA signalization. Thus, the cumulative impact of several VEGFA isoforms depends not only on their absolute expression levels, but also on proportions between particular variants. We therefore decided to estimate the ratios between VEGFA-189, VEGFA-165, and VEGFA-121. Since the amount of other VEGFA variants in examined samples is negligible in comparison with the amount of three predominant isoforms, we used simplified calculation model considering the amount of all mature VEGFA transcripts as a sum of VEGFA-189, VEGFA-165, and VEGFA-121 copy numbers in each individual sample and calculated fractions of each of the predominant isoforms (See Data S4). A significant decrease in the VEGFA-189 fraction and a corresponding increase in fractions of VEGFA-165 and VEGFA-121 were observed in HCC tissue (Figs. 3A–3D). Cluster analysis identified two distinct subsets of cases in which reduction in the VEGFA-189 fraction is complemented by an increase in either the VEGFA-165 or VEGFA-121 fractions (Fig. 3E).

Correlation analysis revealed that changes in VEGFA isoform fractions were stronger associated with clinically significant tumor characteristics than alterations in expression levels (Table 2). Increase in expression levels of VEGFA-spliced, VEGFA-165 or VEGFA-121 correlated with ascites presence and feeble or absent tumor capsule. In contrast, the reduction of VEGFA-189 fraction and the increase in VEGFA-121 fraction were associated with advanced tumor stages estimated using TNM (*Edge et al., 2010*) or BCLC (*Bruix, Reig & Sherman, 2016*) systems. A decrease in the VEGFA-189 fraction was also associated with poor HCC differentiation and a higher level of serum AFP.

## DISCUSSION

There is growing evidence that aberrations in alternative splicing play significant role in carcinogenesis. Splice isoforms may possess different functional properties defining their pro- or anti-oncogenic activity (*Ladomery, 2013*; *Berasain et al., 2014*; *Elizalde et al., 2014*).

Spectrum and ratios of expressed VEGFA isoforms are tissue-specific and provide the establishment of vascular network matching specific tissue functions (*Ng et al., 2001*;

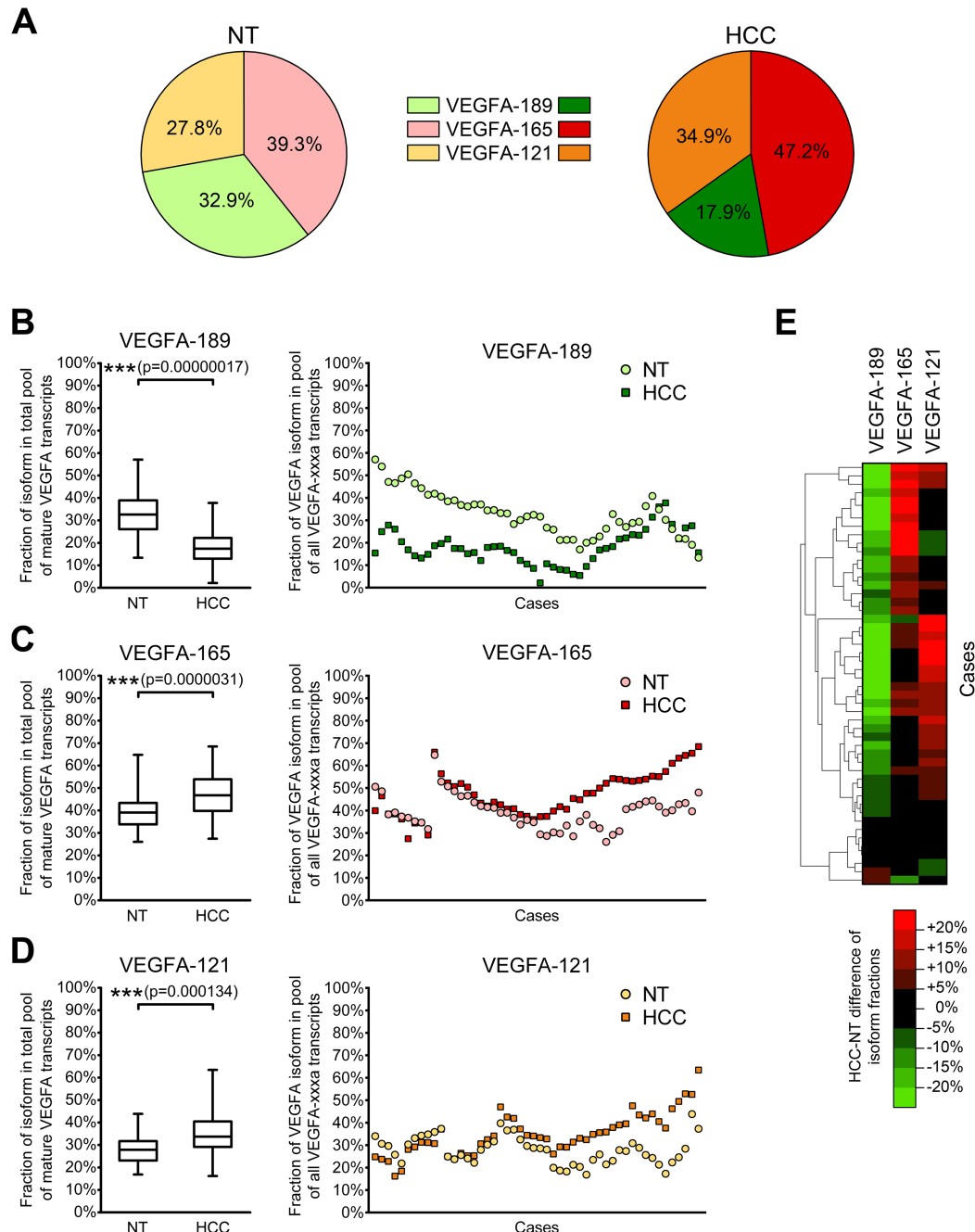

**Figure 3 Alterations of fractions of VEGFA-189, VEGFA-165, and VEGFA-121 in the total pool of mature VEGFA transcripts.** (A) Average proportions of major VEGFA variants in HCC and NT tissue. (B–D) Changes in fractions of VEGFA-189 (B), VEGFA-165 (C), and VEGFA-121 (D) in the total pool of mature VEGFA transcripts. The order of cases presented on dot-plots is rearranged for each isoform to better reflect change trends. (E) Cluster analysis of changes in VEGFA fractions. Data for whole NT and HCC sample sets ($n$ = 50) are presented as box-and-whiskers plots of isoform fractions, data for individual cases are presented as isoform fractions (dot-plots) or differences between paired HCC and NT samples in linear scale (heatmap plot). NS—non-significant difference.

**Table 2 Correlations between HCC/NT ratios of expression levels or fractions of VEGFA isoforms and tumor clinicopathological properties.**

| | Expression level fold change | | | | Isoform fraction fold change | | |
|---|---|---|---|---|---|---|---|
| | VEGFA-spliced | VEGFA-189 | VEGFA-165 | VEGFA-121 | VEGFA-189 | VEGFA-165 | VEGFA-121 |
| Gender | 0.007 (NS[a]) | −0.013 (NS) | 0.068 (NS) | −0.001 (NS) | 0.024 (NS) | 0.096 (NS) | 0.085 (NS) |
| Age | −0.037 (NS) | −0.078 (NS) | −0.030 (NS) | 0.087 (NS) | −0.064 (NS) | −0.164 (NS) | 0.168 (NS) |
| TNM tumor stage | 0.140 (NS) | −0.034 (NS) | 0.194 (NS) | 0.207 (NS) | **−0.297 (0.036)** | −0.070 (NS) | 0.238 (NS) |
| BCLC tumor stage | 0.100 (NS) | −0.084 (NS) | 0.141 (NS) | 0.229 (NS) | **−0.328 (0.020)** | −0.082 (NS) | **0.286 (0.044)** |
| Primary tumor size | −0.004 (NS) | −0.099 (NS) | 0.002 (NS) | 0.053 (NS) | −0.154 (NS) | 0.123 (NS) | 0.103 (NS) |
| Intrahepatic metastases presence | 0.037 (NS) | −0.034 (NS) | 0.151 (NS) | 0.157 (NS) | −0.142 (NS) | 0.019 (NS) | 0.161 (NS) |
| Lymph node metastases presence | 0.103 (NS) | 0.121 (NS) | 0.157 (NS) | 0.153 (NS) | −0.081 (NS) | **−0.348 (0.013)** | 0.247 (NS) |
| Distant metastases presence | −0.209 (NS) | −0.199 (NS) | −0.066 (NS) | −0.230 (NS) | −0.087 (NS) | 0.020 (NS) | −0.153 (NS) |
| Tumor capsule presence | −0.241 (NS) | −0.097 (NS) | **−0.359 (0.018)** | −0.231 (NS) | 0.160 (NS) | −0.208 (NS) | 0.020 (NS) |
| Invasion into blood vessels | 0.161 (NS) | 0.060 (NS) | 0.161 (NS) | 0.265 (NS) | −0.091 (NS) | −0.077 (NS) | 0.204 (NS) |
| Tumor vascularity | 0.123 (NS) | 0.309 (NS) | 0.125 (NS) | 0.010 (NS) | 0.161 (NS) | −0.142 (NS) | −0.212 (NS) |
| Edmondson-Steiner differentiation grade | 0.012 (NS) | −0.168 (NS) | 0.064 (NS) | 0.023 (NS) | **−0.397 (0.016)** | 0.152 (NS) | 0.084 (NS) |
| Serum AFP level | 0.050 (NS) | −0.261 (NS) | 0.067 (NS) | 0.200 (NS) | **−0.420 (0.003)** | 0.248 (NS) | 0.219 (NS) |
| Ascites presence | **0.336 (0.017)** | 0.231 (NS) | **0.324 (0.022)** | **0.318 (0.024)** | 0.067 (NS) | −0.120 (NS) | 0.108 (NS) |
| Cirrhosis presence | 0.054 (NS) | −0.034 (NS) | 0.082 (NS) | 0.230 (NS) | −0.094 (NS) | −0.066 (NS) | 0.162 (NS) |
| Tumor necrosis presence | 0.006 (NS) | −0.118 (NS) | −0.062 (NS) | 0.021 (NS) | −0.018 (NS) | 0.086 (NS) | 0.024 (NS) |

**Notes:**
Data presented as Spearman's correlation coefficients (*p*-value). Statistically significant values are displayed in bold text.
[a] NS—non-significant correlation.

*Vempati, Popel & Mac Gabhann, 2014*). The difference in physiological availability and receptor interactions of alternative VEGFA variants modulates their angiogenesis-stimulating potential (*Plouët et al., 1997*; *Ferrara, Gerber & LeCouter, 2003*; *Rapisarda & Melillo, 2012*; *Vempati, Popel & Mac Gabhann, 2014*). VEGFA is an important regulator of HCC progression and was previously reported to be overexpressed in HCC tissue, so aberrations in isoform expression balance may directly affect tumor development (*Tseng et al., 2008*; *Shen, Hsu & Cheng, 2010*; *Chekhonin et al., 2013*). However, most studies regarding the role of VEGFA in HCC development estimate only total expression level, while the data on VEGFA isoforms expression in liver tumors is still incomplete and contradictory. In the present study, we used paired HCC and NT tissue samples to quantitatively evaluate tumor-specific changes in expression levels of VEGFA isoforms, ratios between them and correlations of these changes with clinically relevant tumor properties.

While VEGFA is known to be frequently overexpressed in various tumors (*Ferrara, Gerber & LeCouter, 2003*; *Chekhonin et al., 2013*), our results indicate a considerable heterogeneity of total VEGFA expression alterations in HCC. Similar results based on immunohistochemical staining were reported (*Tseng et al., 2008*) and explain the lack of significant difference in VEGFA-spliced expression between whole NT and HCC sample sets.

Anti-angiogenic VEGFA-xxxb variants, which are predominant in several normal tissues (*Harper & Bates, 2008*; *Varey et al., 2008*), comprise a very small fraction of all VEGFA transcripts in examined samples (Fig. 2B). Anti-angiogenic action of VEGFA-xxxb variants is supposed to be carried out by competing with VEGFA-xxxa for binding to VEGF receptors. Since the VEGFA-xxxb fraction is negligibly small not only in HCC, but also in NT tissue, we suppose that VEGFA-xxxb isoforms do not exert significant impact upon VEGFA signaling in the liver.

In agreement with previously published data, VEGFA-189, VEGFA-165, and VEGFA-121 isoforms were the major variants expressed in non-cancerous liver (*Ng et al., 2001*; *Sheen et al., 2005*; *Li et al., 2006*; *Iavarone et al., 2007*). All three variants were reported to be overexpressed in HCC tissue compared to independent (VEGFA-189, VEGFA-165, VEGFA-121) or paired (VEGFA-165) NT specimens (*Jeng et al., 2004*; *Li et al., 2006*; *Iavarone et al., 2007*). Surprisingly, most HCC samples examined in the present study displayed decrease in VEGFA-189 expression, while VEGFA-165 and VEGFA-121 variants could be either up- or downregulated (Figs. 2C–2E). We attribute this inconsistence with previous studies to different etiology of HCC samples used in present study (paired samples, hepatitis-negative cases only), the usage of isoform-specific primers instead of universal ones and limitations of semi-quantitative conventional RT-PCR approach used in previous studies. This is the first time VEGFA-189 is demonstrated to be repressed in the majority of HCC cases indicating that its role in HCC progression may be significantly different from that of VEGFA-165 and VEGFA-121.

As stated above, the activity of VEGFA-induced signaling can be affected not only by changes in individual isoforms expression, but also by changes in their balance. The shift of the isoform ratio from poorly diffusible VEGFA-189 variants towards more angiogenic VEGFA-165 and VEGFA-121 ones observed in HCC tissue is much more prominent than alterations of expression levels of corresponding isoforms. There are two possible causes of VEGFA-189 fraction reduction: direct VEGFA-189 downregulation or increase in VEGFA-165 and VEGFA-121 expression levels. According to our results, these two alterations rarely occur simultaneously (Fig. 2F), thus implying that mechanisms controlling VEGFA-189 expression may be considerably different from ones regulating expression of VEGFA-165 and VEGFA-121. This hypothesis is further supported by the fact that reduction in VEGFA-189 fraction is accompanied by increase in fraction of either VEGFA-165 or VEGFA-121 but rarely both (Fig. 3E). Our data indicate that splicing of VEGFA pre-mRNA is controlled by a very elaborate mechanism. Supposedly, HCC development is accompanied by deregulation of this mechanism that results in VEGFA pre-mRNA being preferably processed into VEGFA-165 and VEGFA-121 variants instead of VEGFA-189, but available information on VEGFA splicing mechanisms is insufficient for making more explicit statement.

The importance of VEGFA isoforms fractions evaluation is clearly illustrated by the fact that changes in VEGFA isoform fractions, but not in their expression levels, are associated with such essential HCC clinicopathological features as TNM and BCLC stages, the latter currently being considered the most effective system for HCC prognosis and treatment optimization (*Edge et al., 2010*; *Bruix, Reig & Sherman, 2016*). Moreover,

prominent reduction of VEGFA-189 fraction is associated with more aggressive tumor phenotype (advanced TNM and BCLC stages, poor differentiation level and higher serum AFP level) further supporting our hypothesis of distinct role of VEGFA-189 downregulation in HCC progression and implying possible tumor-suppressive functions of VEGFA-189.

Increase in VEGFA-165 and VEGFA-121 expression levels is associated with ascites and feeble tumor capsule indicating their progression-stimulating role consistent with previously reported data (*Sheen et al., 2005*). High levels of pro-angiogenic VEGFA isoforms expression may contribute to sensitivity of tumor to anti-VEGFA drugs like bevacizumab (*Bates et al., 2012*). However, VEGFA-121 overexpression can reduce tumor sensitivity to inhibitors of KDR receptor, particularly sorafenib, since VEGFA-121 mainly acts through alternative FLT-1 receptor (*Rapisarda & Melillo, 2012*; *Vempati, Popel & Mac Gabhann, 2014*).

Data on VEGFA-189 functions in different tumors are rather controversial. This isoform was reported to be overexpressed in colon, ovary and lung tumors, while its repression was described in non-small cell lung carcinoma (*Vempati, Popel & Mac Gabhann, 2014*). On the one hand, VEGFA-189, like other VEGFA-xxxa isoforms, may exert pro-oncologic effects. Its overexpression can promote migration of breast cancer cells and is associated with colon cancer and lung cancer metastases (*Tokunaga et al., 1998*; *Nishi et al., 2005*; *Hervé et al., 2008*). VEGFA-189 stimulates the growth of colon tumors in vivo, but to a lesser extent than VEGFA-165 (*Tomii et al., 2002*). On the other hand, in breast cancer VEGFA-189 may possess anti-tumor functions since it reduces invasion and metastatic potential of tumor cells and promotes apoptosis in stress conditions (*Vintonenko et al., 2011*; *Di Benedetto et al., 2015*). It also exerts opposite effects on proliferation of endothelial cells originating from different tissues (*Hervé et al., 2005*). Unlike VEGFA-164 and VEGFA-120, VEGFA-188 (mouse counterparts of human VEGFA-165, VEGFA-121, and VEGFA-189 isoforms, respectively) does not affect fibrosarcoma cells proliferation and migration, but induces apoptosis (*Kanthou et al., 2014*). It is likely that VEGFA-189 functions are tissue-specific and can differ considerably from those of VEGFA-165 and VEGFA-121 since unprocessed VEGFA-189 is much weaker stimulator of angiogenesis and, possibly, acts via distinct signaling pathways (*Plouët et al., 1997*; *Vempati, Popel & Mac Gabhann, 2014*). We suppose that reduction of the VEGFA-189 fraction in HCC tissue can contribute to the development of more aggressive tumor phenotype. Possibly such tumors could be more sensitive to anti-VEGFA treatment like bevacizumab therapy due to increased fractions of VEGFA-165 and VEGFA-121.

Current studies of the role of VEGFA splice variants in cancer development are mainly focused on balance of pro- and anti-angiogenic VEGFA isoform groups. At this time several regulators of VEGFA mRNA splicing have been identified (SRp40, SRp55, SRSF1) that interact with exon 8, thus controlling generation of VEGFA-xxxa or VEGFA-xxxb variants (*Nowak et al., 2008*, *2010*). However, limited information is available on possible regulators of VEGFA mRNA splicing events involving exons 5–7 that result in generation of VEGFA-189, VEGFA-165, and VEGFA-121. SRp20, SRp40, and SRSF1 proteins were reported to be connected to hypoxia-induced shift in VEGFA isoform balance towards

VEGFA-121, but their ability to directly interact with sequence of exons 5–7 is yet to be investigated (*Elias & Dias, 2008*) . VEGFA-xxxa splicing can also be influenced by non-coding RNAs (MALAT1) and chromatin modifiers (EHMT2); the latter can specifically prevent the inclusion of exon 6a into VEGFA mRNA thus shifting the balance from VEGFA-189 towards VEGFA-165 (*Salton, Voss & Misteli, 2014*; *Pruszko et al., 2017*).

Since there is no reliable experimental approach to induce VEGFA splicing changes, most of published reports on functions of VEGFA-xxxa isoforms evaluate effects caused by overexpression of certain VEGFA variants (*Tomii et al., 2002*; *Hervé et al., 2008*; *Vintonenko et al., 2011*; *Kanthou et al., 2014*; *Di Benedetto et al., 2015*). While inactivation of a single VEGFA isoform is much more complicated task, there is only one published investigation describing ribozyme-mediated specific cleavage of VEGFA-189 in non-small cell lung cancer cells that resulted in attenuation of their malignant potential (*Oshika et al., 2000*). Given possible tissue specificity of VEGFA-189 functions, additional experiments on its inactivation are necessary to determine possible clinical impact of VEGFA-189 and its role in tumor development and angiogenesis. Implementation of highly specific RNA interference approaches to selectively knockdown VEGFA-189 expression and identification of VEGFA exon 6 splicing regulators could provide valuable information essential for achieving this goal.

## CONCLUSIONS

Using a quantitative approach, we have detected HCC-specific shift of VEGFA isoforms ratios that consisted in decrease in VEGFA-189 and increase in VEGFA-165 and VEGFA-121 fractions. These changes were associated with multiple essential clinicopathological tumor characteristics. The clinical significance of presented data consists in their potent impact on optimization of HCC treatment since the VEGFA isoforms ratio may be a promising factor for prediction of anti-angiogenic therapy efficiency. Further studies of VEGFA isoforms expression, VEGFA mRNA splicing regulation and VEGFA-189 functional properties in HCC are necessary in order to evaluate its possible association with other tumor progression factors, survival and recurrence.

## ACKNOWLEDGEMENTS

The authors express their gratitude to Prof. Francisco X. Real (Epithelial Carcinogenesis Group, CNIO, Madrid, Spain) for his help in interpreting data generated in RT-qPCR experiments.

### Funding

This study was funded by Russian Ministry of Education and Science (contract 14.607.21.0049, RFMEFI60714X0049). The funders had no role in study design, data collection and analysis, decision to publish, or preparation of the manuscript.

## Grant Disclosures

The following grant information was disclosed by the authors:
Russian Ministry of Education and Science (contract 14.607.21.0049, RFMEFI60714X0049).

## Competing Interests

The authors declare that they have no competing interests.

## Author Contributions

- Mikhail S. Chesnokov conceived and designed the experiments, performed the experiments, analyzed the data, prepared figures and/or tables, authored or reviewed drafts of the paper, approved the final draft.
- Polina A. Khesina conceived and designed the experiments, performed the experiments, analyzed the data, prepared figures and/or tables.
- Darya A. Shavochkina performed the experiments.
- Inna F. Kustova performed the experiments.
- Leonid M. Dyakov performed the experiments.
- Olga V. Morozova performed the experiments, contributed reagents/materials/analysis tools.
- Nikolai S. Mugue performed the experiments, contributed reagents/materials/analysis tools.
- Nikolay E. Kudashkin contributed reagents/materials/analysis tools.
- Ekaterina A. Moroz contributed reagents/materials/analysis tools.
- Yuri I. Patyutko contributed reagents/materials/analysis tools.
- Natalia L. Lazarevich conceived and designed the experiments, analyzed the data, prepared figures and/or tables, authored or reviewed drafts of the paper, approved the final draft.

## Human Ethics

The following information was supplied relating to ethical approvals (i.e., approving body and any reference numbers):

All procedures performed were approved by the medical ethics committee of FSBI "N.N. Blokhin National Medical Research Center of Oncology"of the Ministry of Health of the Russian Federation.

## Data Availability

The research in this article did not generate any new raw sequence data besides sequences that were generated to confirm that the PCR products described in the article correspond to certain VEGFA isoforms. These sequences are present in Table S2.

## Supplemental Information

Supplemental information for this article can be found online at http://dx.doi.org/10.7717/peerj.4915#supplemental-information.

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
