# Peer review of "Shift in VEGFA isoform balance towards more angiogenic variants is associated with tumor stage and differentiation of human hepatocellular carcinoma"

_PeerJ, doi:10.7717/peerj.4915_

## Round 0.1 · original submission · Minor Revisions

Please carefully modified the text and presentation as suggested by reviewer#1. As suggested by reviewer#2, please perform a Western blot analysis to semi-quantitatively analyze VEGFA isoforms and correlate with the HCC properties. Provide more details on the characterization and validation of the tumor tissue in the materials and methods section as suggested by reviewer#3. Reviewer #3 also suggested to demonstrated how VEGF-A189 specifically is regulated, therefore, either please discuss as future experiments will be done or provide evidence if you can. Clearly describe the impact of your findings in the manuscript.

Reviewer 1 ·

Basic reporting

Article has clear structure, provided with literature references, conclusions are supported with the data.
Most of the raw data shared with the exception of clinicopathological data for each patient - instead, authors summarize those feature in Table 1.

Experimental design

Research questions are clearly defined: authors sought to analyze distribution of different isoforms of VEGFA in HCC and adjacent non-tumor tissues. Elaborated methods were used to quantify the transcripts, those methods are described with sufficient details.

Validity of the findings

Findings are explained and supported by the data. Statistical analysis could have more details (see comments below).

Additional comments

I only have certain minor comments regarding the presentation of the results and methods:
1) Description of primers in Table S1: it may be a bit confusing for the reader that the same primers are shown under different names (for example - VEGFA-total, VEGFA-intron5, VEGFA-189, VEGFA-165, VEGFA-121- is all the same primer).
2) Figure 1A - dashed lines continuing exon 8b might imply presence of other exons which is not the case
3) Figure S1 - for easier understanding, might be useful to mark the heteroduplex band as well
4) Figure 2B - on the left plot, maybe worth showing VEGFA-xxxb distribution for both tissues (like on the right plot)
5) For Table S2, authors should give better explanation for what exactly is shown in the "Unique sequence region" of the table. The sequence provided there is shorter than the length indicated in "PCR product" column (for example, for the row marked "VEGFA-iso (523 bp)" the sequence is 228 bp only, etc.) - I guess it's probably limitation of sequencing - but this should be explained
6) Methods and data of Table 2 should be better explained. For example, what measure was used for the "change" - fold change, log2, etc? Which variables were used as continuous, categorical, or discreet? As authors show Spearman's correlation, I assume all the data was ranked - but this should be explained explicitly.
7) To visualize correlation with clinicopathological properties, it could be helpful to add corresponding color bands on the heatmaps (Fig. 2D, Fig. 3C)
8) Right barplots at Fig. 3B might be confusing (one would assume to sum up values within one bar which is not the case). As one alternative, authors could use a scatterplot instead: each case would be represented by a dot with each axis corresponding to a tissue type.
9) Authors performed valuable work to determine absolute number of copies for each transcript. It would be interesting to estimate approximate number of copies per cell or at least per mg of tissue: this would give reference potentially helpful for future researchers.
10) In contrast to literature data about other tissues, authors did not find significant VEGFA-xxxb expression in both normal and tumor liver tissues - it's probably worth to mention in the abstract.

·

Basic reporting

The manuscript "Shift in VEGFA isoform balance towards more angiogenic variants is associated with tumor stage and differentiation of human hepatocellular carcinoma" is well written in plain and professional English. It provides a short, but sufficient overview of the background, with relevant references.
The figures are of sufficient quality, mostly quantification of the raw data (which is also provided).
The hypothesis is clearly formulated and addresses an important issue of different roles of splice variants in regulation of VEGFA role in HC progression. In addressing this issue, clear questions are answered with robust results.

Experimental design

While the semi-quantitative and quantitative analysis of the VEGFA isoforms is well done, the analysis is limited to the RNA level. This is problematic for 2 reasons:
1) There is no corroboration of the data with independent methods. This may result in biasing due to the chosen methodology.
2) It is well known that VEGFA in particular is regulated very extensively at all levels of expression: splicing, transcription AND translation (for review see Nucleic Acids Res. 2013 Sep; 41(17): 7997–8010). Moreover, VEGFA is active at a protein level rather than at the mRNA level.
Therefore, I suggest the authors confirm the results they obtained with a simple semi-quantitative Western blot analyis. They may use total VEGFA antibodies and distinguish isoforms with SDS-PAGE electrophoresis by size, which is a trivial task. It is important to confirm close correlation between the qPCR and the Western blot data, as well as the correlation with the tumor characteristics (see below).
Otherwise, the obtained results are quantified and analyzed well.

Validity of the findings

The most important finding of the article is shown in table 2. It gives validity to the data obtained, confirms the original hypothesis and even provides some predictability power to the obtained results on VEGFA isoforms.
It is important to notice, however, that the best obtained correlation coefficients and the p-values are not very strong. They are within the 0.05 standard (for the p-values), but in all but one cases higher than 0.01, for instance. This is especially critical because the authors are analyzing their data for correlation against 16 tumor sample characteristics, so a possibility of finding a low statistically significant correlation by chance is high.
The data can be strengthened if the authors, again, provide a similar analysis using quantification on a protein level. This would be an independent analysis, which, together with the already produced data, could make it more robust and biologically relevant.

Additional comments

In general, I like the article for its logical structure, sound data and clinical relevance. However, in my opinion the data needs to be corroborated with an independent method and on another level of gene expression. Therefore, I suggest to use A Western blot analysis to semi-quantitatively analyze VEGFA isoforms and correlate with the HCC properties.

·

Basic reporting

The language is clear and professional throughout the manuscript, litterature references of relevance to the topic are provided, and the background information presented is sufficient to provide context. The structures is professional and the article is self-contained.

Experimental design

The research question is well defined, and the statistical Foundation for the study is solid. It would have been easier to interpret the expression levels of VEGF-A isoforms if a better description of the samples had been given. How did the authors know that the samples were in fact tumor or non-tumor samples (this can be very difficult to judge from simply looking macroscopically at the tissue). What extent of tumor cells versus non-tumor cells were present in the tumor tissue? What extent of necrosis/viable cells was present in the tumor tissue? The selection, characterization and validation of the tumor tissue should be described in more detail in the materials and methods section.
We are asked to judge whether the research fills an identified knowledge gap. It is known that VEGF-A is deregulated in many tumors, and the authors also make this clear in both the introduction and discussion. Is it a knowledge gap how VEGF-A189 specifically is regulated? Unless the authors can demonstrate that the down-regulation of VEGF-A189 without the up-regulation of other VEGF-isoforms, as what the authors are demonstrating in their work, leads to a difference in angiogenesis, it is hard to see how the results fill a knowledge gap. That does not mean, however, that the results could not be proven to do so in the future.

Validity of the findings

The impact of the findings are not clear at present, as the functional consequence of reduced VEGF-A189 levels are not known. The data, however, seem robust and statistically sound, and does seem to impact tumor progression (or advanced progression may impact VEGF-A189 production). The conclusions and discussions are however done at a reasonable level, without attempts to overreach.

---

## Round 0.2 · accepted · Accept

Your revised manuscript is acceptable in current form.

#